# Analysis of ^16^O/^18^O and H/D Exchange Reactions between Carbohydrates and Heavy Water Using High-Resolution Mass Spectrometry

**DOI:** 10.3390/ijms23073585

**Published:** 2022-03-25

**Authors:** Lidiia Rumiantseva, Sergey Osipenko, Artem Zharikov, Albert Kireev, Evgeny N. Nikolaev, Yury Kostyukevich

**Affiliations:** Skolkovo Institute of Science and Technology, 121205 Moscow, Russia; lidiia.rumiantseva@skoltech.ru (L.R.); sergey.osipenko@skoltech.ru (S.O.); a.zharikov@skoltech.ru (A.Z.); a.kireev@skoltech.ru (A.K.); e.nikolaev@skoltech.ru (E.N.N.)

**Keywords:** carbohydrates, mass spectrometry, H/Dexchange, ^16^O/^18^Oexchange reaction

## Abstract

Mono- and polysaccharides are an essential part of every biological system. Identifying underivatized carbohydrates using mass spectrometry is still a challenge because carbohydrates have a low capacity for ionization. Normally, the intensities of protonated carbohydrates are relatively low, and in order to increase the corresponding peak height, researchers add Na^+^, K^+^, or NH_4_^+^to the solution. However, the fragmentation spectra of the corresponding ions are very poor. Based on this, reliably identifying carbohydrates in complex natural and biological objects can benefit frommeasuring additional molecular descriptors, especially those directly connected to the molecular structure. Previously, we reported that the application of the isotope exchange approach (H/D and ^16^O/^18^O) to high-resolution mass spectrometry can increase the reliability of identifying drug-like compounds. Carbohydrates possess many –OH and –COOH groups, making it reasonable to expect that the isotope exchange approach would have considerable potential for detecting carbohydrates. Here, we used a collection of standard carbohydrates to investigate the isotope exchange reaction (H/D and ^16^O/^18^O) in carbohydrates and estimate its analytical applications.

## 1. Introduction

Carbohydrates are molecules that are widely distributed in biological systems. They take part in many biological processes as separate molecules or as parts of complex macromolecules. Carbohydrates are the primary energy source for most cells that perform a structural function or that areattached to proteins during the glycosylation process. Moreover, carbohydrates are involved in cell–cell recognition processes [1] and disease progression [2]. Due to the important role of this class of compounds, it is often necessary to know the carbohydrate content in complex biological samples. As mass spectrometry is the key instrument used in content analysis of biological samples, efforts should be made to solve the problemof using it to detect and precisely identify carbohydrates.

Identifying carbohydrates using mass spectrometry is a very challenging issue. The first reason is that carbohydrates are hardly detected by the ESI/MS method, especially in low concentrations. In contrast to the ESI of proteins, the ESI of carbohydrates mainly generates cationized ions instead of protonated ions. The intensity of these cationized ions is usually much lower than the intensity of protonated peptide ions. It was shown that the intensity of sodiated carbohydrate ions could be increased 10–50 times by adding a small amount of salt to the solution [3]. At the same time, protonated ions of carbohydrates might not be obtained at all. That is why preliminary derivatization and separation by gas chromatography are used. However, derivatization methods such as methylation—usually used for exploring the carbohydrates in a sample—significantly complicate the process of sample preparation. That is why it is reasonable to propose other methods foridentifying carbohydrates.

Another reason to use mass spectrometry to identify carbohydrates is that, for even the simplest molecular formulas ofcarbohydrates, there are several structural and stereoisomers. That is why every molecular descriptor that could help define the structural formula is very important. Especially meaningful are descriptors which can be accurately predicted referring only to the structural formula of the molecule independently ofexperimental conditions. Different selective reactions might be used to obtainsuch descriptors. In this case, the mass shift after the reaction corresponds to the number of particular chemical groups in the molecule and can be used to accurately count the descriptors.

Hydrogen/deuterium(H/D) andisotope (^16^O/^18^O) exchange may be considered reactions that providestructural insights about particular functional groups, as they preserve most molecular properties that are important for analysis. For instance, introducing isotope labels typically maintains chromatographic retention times, making data processing simpler. The first mention of the ^16^O/^18^Oexchange reaction appeared in the 1930s [4]. Later, the S_N_1 and S_N_2 mechanisms, along with rates [5] of the reaction, were presented for molecules with various functional groups. The ^16^O/^18^Oexchange reaction takes hours and requires acidic catalysis and high temperatures. It was shown that oxygen atoms of carboxyl and carbonyl groups undergo this reaction under mild conditions. However, the exchange in other functional groups can be achieved by conducting the reaction under stricter conditions. The H/Dexchange reaction is much faster, and it occurs immediately after dissolving the sample in heavy water (D_2_O). The primary exchangeable groups are −OH, −COOH, −NH, and −SH. However, it is possible to achieve the exchange in electron-enriched sections of the aromatic ring by using high temperature or acid/base catalysis [6].

Each oxygen or hydrogen exchange leads to the shifting of masses of molecular or fragment ions where this exchange occurs. These shifts correspond to the differences between masses of ^18^O and ^16^O, or H and D:∆Moxygen=M(O18)−M(O16)=17.99916Da−15.99491Da =2.00425Da
$$∆M_{hydrogen}=M\left(D\right)-M\left(H\right)=2.01410\mathrm{Da}-1.00783\mathrm{Da}=1.00627\mathrm{Da}$$

Isotope exchange reactions are widely used for different aims during biological andchemical experiments. The most known biological research using stable isotopes is on the mechanism of photosynthesis [7] and ondeterminingthe right inheritance-holding molecules [8]. The stable oxygen isotope ^18^O is used to increase the reliability [9,10,11] of compound identification and to track some metabolic pathways [9]. The stable hydrogen isotope D is also used toincreasethe reliability [6] of compoundidentification, as well as investigate protein structure [10]. Since heavy water (containing ^18^O or D instead of ^16^O or H) became generally available, much research has beenconducted in the field of exchange reactions between heavy water and organic compounds [11].

Much research was dedicated to measuring the isotope ratio (^18^O/^16^O) of sugars and their derivatives [12,13]. This information is important in investigating biosynthesis [14,15,16] and geochemical processes. However, to our knowledge, the ^16^O/^18^Oexchange reaction has not been used as an analytical method for detecting carbohydrates.

Application of the hydrogen/deuterium exchange for investigating carbohydrates was previously reported [17,18,19,20]. E. Gallagher showed that it was possible to distinguish three different carbohydrate isomers using the H/D exchange reaction [21]. We previously reported the observation of bimodal deuterium distribution, depending on the solvent used for protonated, deprotonated, and cationized carbohydrates [22,23]. Such an effect was explained by the presence of two conformations of carbohydrate ions—“closed”, in which the exchange was hindered due to the steric obstacles, and “open”, in which the exchange occurredquickly. 

The H/Dand ^16^O/^18^Oexchangereactions should be investigated on individual compounds prior to using these approachesso as to identify carbohydrates in complex biological samples. In this paper, we present basic research on the capacity of individual carbohydrates to undergo the H/Dand ^16^O/^18^Oexchange reactions.

## 2. Results

### 2.1. Products of Oxygen Isotope Exchange Reaction of Monosaccharides

Monosaccharides are mainly present in solutions in several cyclic forms, which may interconvert through an open-chain form. Since an open-chain form contains a carbonyl group, it is expected to be involved in ^16^O/^18^Oexchange reactions in heavy-isotope-enriched media, providing a monosaccharide with one ^18^O label. Indeed, it was previously demonstrated that monosaccharides might exchange one carbonyl oxygen atom. In our previous investigations of ^16^O/^18^O exchange reactions, we found that hydroxylgroups werelikely to exchange only in electron-deficient positions, such as allyl or benzyl.

Surprisingly, our experiments showed an additional exchange in most examined monosaccharides after incubation at 95 °C for 48 h. Figure 1A–F demonstrates the mass spectraof monosaccharides before and after incubation. Two shifts of a sodiated ion by 2.004 were observed for D-glucose, D-mannose, D-rhamnose, D-xylose, D-arabinose, and D-galactose. 

This additional exchange may occur in hydroxyl groups of monosaccharides. However, we could notjustify this scenario since these groups have no reasons to be attacked by nucleophiles. We did not observe ^16^O/^18^O exchange in mannitol (Figure 1G), which confirmedthe different exchange mechanisms. The most probable explanation for the additional exchange was the aldose–ketose transformation of monosaccharides (Figure 2). Although such transformations of monosaccharides wereinvestigatedwell, we suppose that ^16^O/^18^O exchange may become an instrument for investigating transformations in oligosaccharides and polysaccharides. 

For D-glucose, we obtained a collision-induced dissociation spectrum for ions produced in negative ESI mode. Our results are shown in Figure 3. The precursor ion with *m/z* = 181 (with one^18^O label) was isolated with an isolation window of 0.4 Da, ensuring that ions without ^18^O labels were not fragmented. We observed that fragment peak *m/z* = 119 did notshift, the peak with *m/z* = 113 shiftedcompletely by 2.0042 Da, and other peaks demonstrated the presence of both shifted and non-shifted peaks. Analysis of the fragmentation products of isotopically labeled compounds may help in understanding the collision-induced fragmentation mechanism and pathways. It may also help in identifyingcompounds. Previously, we demonstrated how combining the isotope exchange reaction (H/D and ^16^O/^18^O)and LC-MS/MS couldhelp identify compounds in complex biological mixtures [24,25].

Recently, we created a PyFragMS [25] program—a web tool consisting of a database of annotated MS/MS spectra of isotopically labeled molecules (after H/D and ^16^O/^18^O exchange), instruments for investigating fragmentation pathways (creating the fragmentation tree) for arbitrary molecules, and tools for identifying unknown molecules using isotope exchange information and MS/MS data (see https://pyfragms.anvil.app/, accessed date: 22 February 2022). Here, we used this software to investigate the fragmentation pathway of deprotonated D-glucose by considering CID spectra after^16^O/^18^O exchange. 

Previously, the majority of studieswere dedicated to using tandem mass spectrometry toinvestigate oligosaccharides. However, less attention has been paid to the CID spectra of monosaccharides and their derivatives [26,27]. In Figure 4,we present the computed fragmentation tree for 18-labeled D-glucose. A fragmentation tree is a representative graph in which two product ions are connected with an edge only if they can be directly attributed to a single fragmentation event. Fragmentation itself is generally a multistep process, and a peak observed in the spectrum can be a product of a series of fragmentation events.

We can observe different possible fragmentation pathways. Unlabeled peak *m/z* = 119 is a result of the bond cleavage between C1–C2 and O6–C5. The atoms are numbered, as shown in Figure 4.This numeration corresponds to the atom’sserial number in the canonical SMILES notation representation of D-glucose—C(C1C(C(C(C(O1)O)O)O)O)O. The unlabeled peak *m/z* = 89 couldbe formed as a result ofthe fragmentation of the precursor ion or of the fragment ion, *m/z* = 119. Two different pathways of fragmentation are possible. Pathways for the formation of labeled peaks are also shown. It must be emphasized that labeled peak *m/z* = 73 is formed via fragmentation of the labeled peak *m/z* = 91, while unlabeled peak *m/z* = 71 forms via direct fragmentation of the precursor.

This discussion of the fragmentation pathways and computed fragmentation trees has the following shortcomings: we didnot consider possible intramolecular rearrangements, nor didwe take into account the possible different sites of deprotonation (currently, we considered deprotonation at the O11 position). To more accurately consider these fragmentation pathways, further research, taking these shortcomings into account, is required.

### 2.2. Products of Oxygen Isotope Exchange Reaction of Oligosaccharides

Considering that ^16^O/^18^O exchange may only occur in carbonyl groups of carbohydrates, linear oligosaccharides should produce molecules with 0–2 heavy isotope labels. This productiondepends on the ability of the end monomers to reach an open-chain configuration. For example, stachyose does not exchange oxygen molecules (Figure 5A). However, in the maltohexaose mass spectrum (Figure 5B), we observed three shifts of the sodiated molecule by 2.004 Da after the isotope exchange reaction. The aldose–ketose transformation of the end monomer may be an explanation for the second exchange in maltohexaose. As maltohexaose is hardly, and irreversibly, hydrolyzed, the only reasonable explanation of the third incorporated ^18^O label is the exchange ofhydroxyl groups.

### 2.3. H/D Exchange Reaction of Carbohydrates

Previously, we observed a bimodal deuterium distribution depending on the solvent used for protonated, deprotonated, and cationized carbohydrates [22]. Such an effect was explained by the presence of two conformations of carbohydrate ions—closed, in which the exchange washindered due to the steric obstacles, and open, in which the exchange occurred quickly. The presence of these twoconformations may complicate the analytical applications of the H/D exchange for the detection of carbohydrates.Furthermore, the configuration of the ion source, which was previously used, is not suited for combination with liquid chromatography. Indeed, previously, the vapors of D_2_O were infused in the ion source by placing a droplet of D_2_O on the metal plate below the desolvating capillary.The evaporation of the droplet resultedin the saturation of the regions around the ESI needle and desolvating capillary with vapors of D_2_O.

Using such a configuration, one may expect that the bimodal distribution of deuterium in the resulting mass spectrum may occur due to the simultaneous exchange of ions in the gas phase (due to the collision with D_2_O molecules) and the liquid phase (due to the penetration of D_2_O into the evaporating droplets). In the present study, we used a different design of the ion source (see Figure 6B). In thisdesign, the vapors of D_2_O are infused directly in the desolvating capillary, minimizing, or even excluding, exchanges in the liquid phase. 

Our results are presented in Figure 7. We still observe the bimodal deuterium distribution, even usingthe novel design of the ion source. This suggests that carbohydrate ions are indeed formed during ESI ionization in twoconformations. 

We also found that the proposed ion source design allowedus to observe a large number of exchanges, making it possible to distinguish carbohydrate ions in complex mixtures. Indeed, when analyzing a mixture of unknown molecules with high molecular mass, it may be difficult to even determine the class of the molecule, especially if the fragmentation pattern of the molecule is poor (as it is in the case of carbohydrates). However, the H/D exchange experiment can determine the number of –OH groups in the molecule, and carbohydrates can be easily distinguished from other classes of molecules by the presence of large numbers of rapidly exchanging hydrogens. The advantage of the proposed method is its independence of ionization type (protonation, deprotonation, or cationization with metal) and the possibility for it to be coupled with liquid chromatography. 

## 3. Materials and Methods

Samples. Ten different compounds were used: D-xylose, D-glucose, D-galactose, D-rhamnose, D-arabinose, D-mannose, mannitol, maltohexaose, stachyose, and dextran.Allthe compounds were purchased from Sigma-Aldrich.

^16^O/^18^O isotope exchange.To perform oxygen isotope (^16^O/^18^O) exchange, we followed the procedure described by Samuel and Silver [5].This method was also used by us in previous studies [28,29]. The target compound was dissolved in H_2_^18^O.The solution was incubated at 95 °C for either 24 or 48 h in a sealed glass vial. Final samples were diluted with methanol to aconcentration of 1 mg/mL to improve ESI spray stability. 

Hydrogen isotope exchange reaction.Hydrogen/deuterium (H/D) exchange was performed in the ESI source using our previously developed approach [24]. The vapors of the D_2_O were infused into the desolvating capillary, heated to 300 °C. The H/D exchange occurredin both the gas phase and—due to the penetration of moisture into the droplets [30]—in the liquid phase. The details of the ion source design, as well as its applicationsfor investigating various compounds, can be found in our previous publications [6].

MS analysis.Mass spectra were acquired on a QExactive Orbitrap system (Thermo Fischer Scientific) with modified MALDI/ESI injector (Spectroglyph, LLC), operated in ESI mode, with 140,000 resolution for both MS and MS/MS spectrameasurements (HCD 10-50 NCE). Solutions of the target compounds in a methanol–water mixture (1:1) were infused with the flow rate 1 µL·min^−1^, and aspray voltage of 3 kV. The spectra were measured for all compounds in positive and negative modes.

As was previously suggested [3], sodium acetate was added to achieve a sufficiently strong signal intensity of the sodiated carbohydrate ions in positive mode.

Data analysis. Spectra were processed using Xcalibur software (Thermo).To check the MS/MSspectra (if available),mzCloud database (https://www.mzcloud.org/, accessed date: 22 February 2022) was used. For the computing fragmentation trees, the PyFragMS software was used (https://pyfragms.anvil.app/, accessed date: 22 February 2022).

## 4. Conclusions

In this study, we presented basic research on the capacityof individual carbohydrates to undergo H/D and ^16^O/^18^Oexchange reactions. We performed the ^16^O/^18^Oexchange reaction to collectmonosaccharides, andobserved the expectedexchange in the carbonyl oxygen (monosaccharides may interconvert through an open-chain form, which contains a carbonyl group). However, when the time of the reaction was extended, we observed an unexpected second exchange occurringin the hydroxyl groups. The most probable explanation for the additional exchange is the aldose–ketose transformation of monosaccharides. 

For D-glucose, after the ^16^O/^18^Oexchange reaction, we obtained the CID fragmentation spectra.By calculating the fragmentation trees using PyFragMS software, we investigated D-glucose’s fragmentation pathways. It was reliably demonstrated that many peaks in the MS/MS spectrum are, in fact, the superposition of product ions of different structures formed via different fragmentation pathways. The^16^O/^18^Oexchange reaction was also observed in the oligosaccharides. We performed a H/D exchange reaction for different oligosaccharides using the ion source design, ensuring the reaction occurredin the gas phase. We observed the bimodal deuterium distribution, proving the presence of twodifferent conformations of gas-phase ions, which differ by the rate of the H/D exchange reaction.

## Figures and Tables

**Figure 1 ijms-23-03585-f001:**
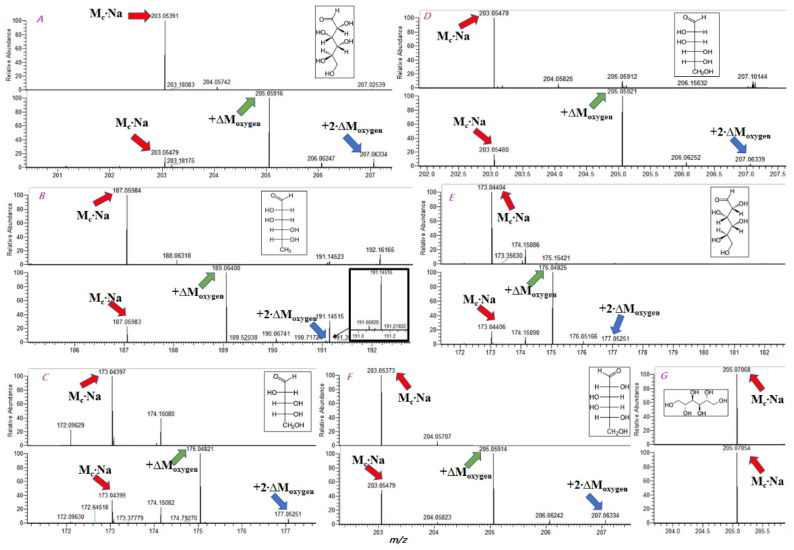
(**A**) Experimental spectrum of D-glucose (positive mode) before (upper) and after (lower) ^16^O/^18^Oexchange reaction. (**B**) Experimental spectrum of D-rhamnose (positive mode) before (upper) and after (lower) ^16^O/^18^Oexchange reaction. (**C**) Experimental spectrum of D-arabinose (positive mode) before (upper) and after (lower) ^16^O/^18^Oexchange reaction. (**D**) Experimental spectrum of D-mannose (positive mode) before (upper) and after (lower) ^16^O/^18^Oexchange reaction. (**E**) Experimental spectrum of D-xylose (positive mode) before (upper) and after (lower) ^16^O/^18^Oexchange reaction. (**F**) Experimental spectrum of D-galactose (positive mode) before (upper) and after (lower) ^16^O/^18^Oexchange reaction. (**G**) Experimental spectrum of mannitol (positive mode) before (upper) and after (lower) ^16^O/^18^Oexchange reaction. Top subfigure indicates no exchange. Bottom subfigure indicates after exchange.

**Figure 2 ijms-23-03585-f002:**
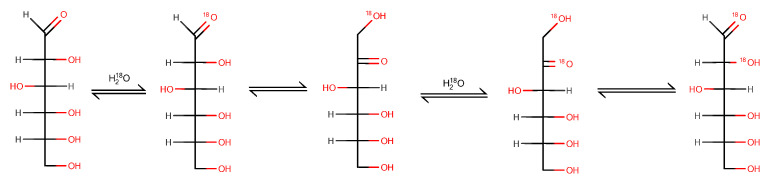
A possible explanation of additional ^16^O/^18^O exchange using D-glucose as an example.

**Figure 3 ijms-23-03585-f003:**
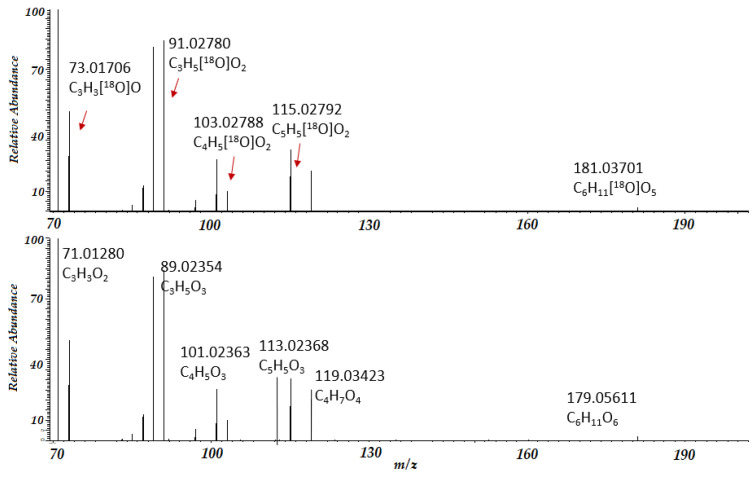
The CID fragmentation spectrum of D-glucose in negative ESI mode, after (**top**) and before (**bottom**) ^16^O/^18^O exchange.

**Figure 4 ijms-23-03585-f004:**
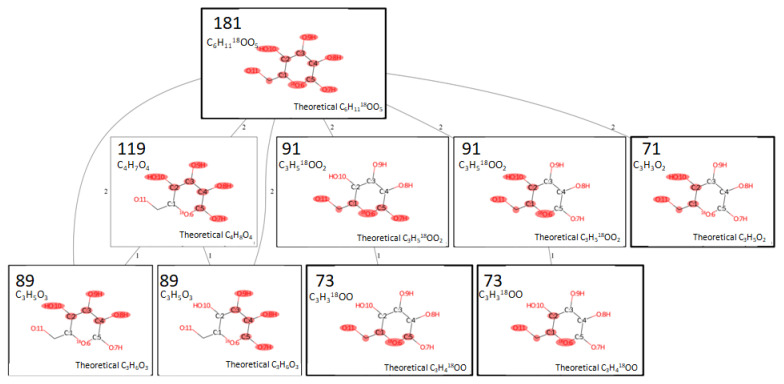
The fragmentation tree obtained for D-glucose after ^16^O/^18^Oexchange reaction. Numbers correspond to the integer *m*/*z* value.The highlighted groups are retained in the fragment ion.

**Figure 5 ijms-23-03585-f005:**
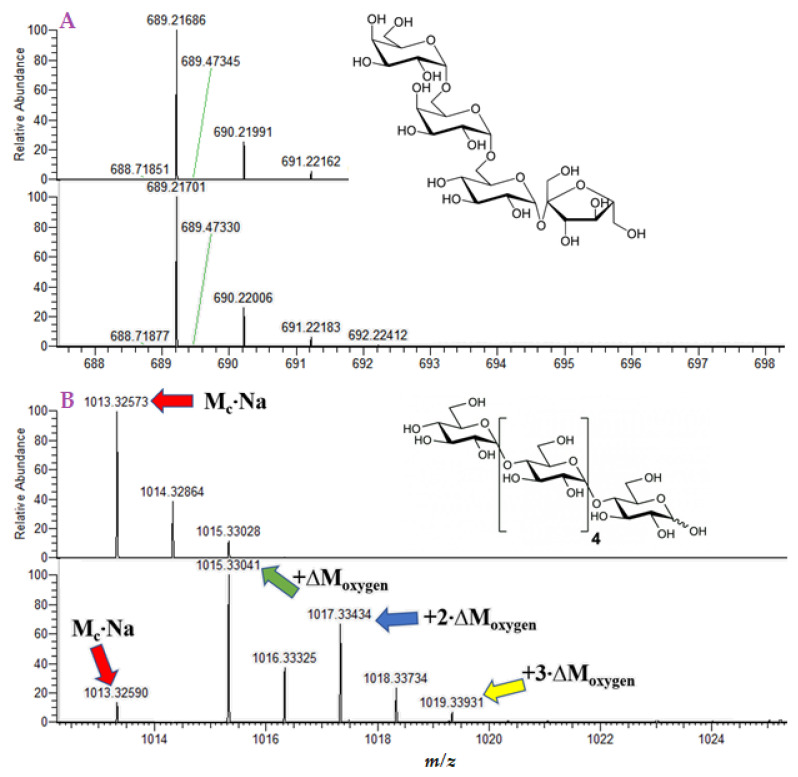
(**A**) Mass spectra of stachyose, before and after ^16^O/^18^O exchange. (**B**) Mass spectra of maltohexaose, before and after ^16^O/^18^O exchange.

**Figure 6 ijms-23-03585-f006:**
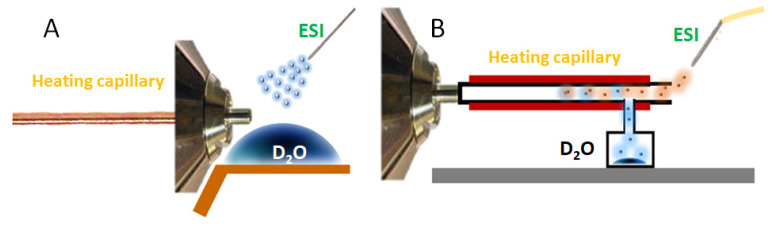
The configuration of the ion sources for performing H/D exchange reactions. (**A**)Previously used configuration, and (**B**)the configuration used in the present paper.

**Figure 7 ijms-23-03585-f007:**
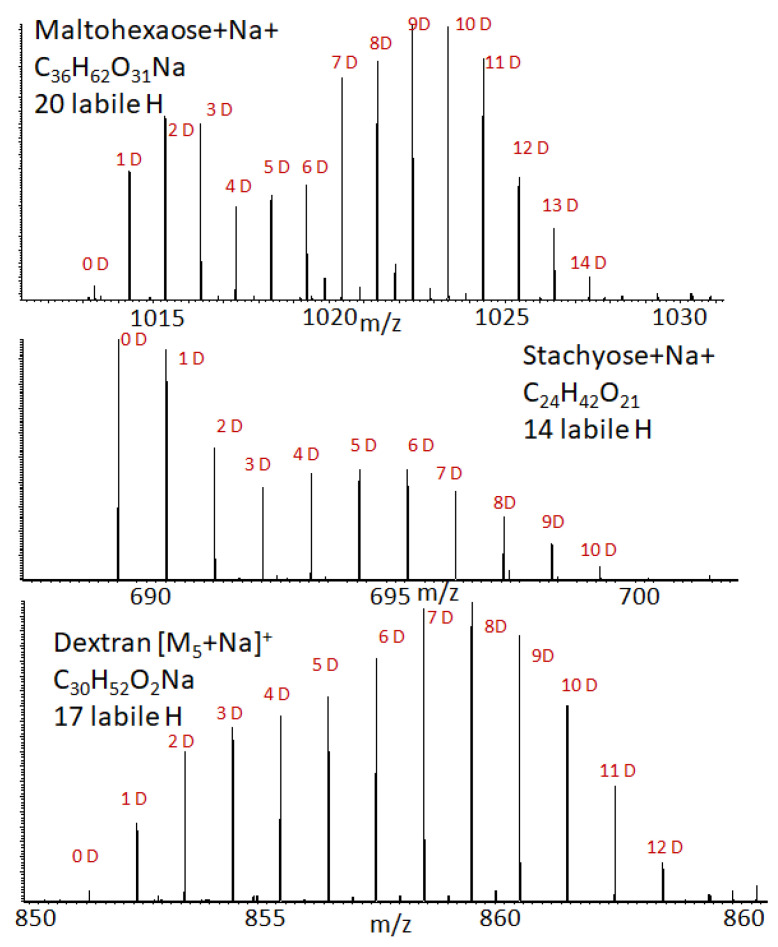
Observation of the bimodal deuterium distribution for selected carbohydrates.

## Data Availability

Not applicable.

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
