# Peer review of "Analysis of 16O/18O and H/D Exchange Reactions between Carbohydrates and Heavy Water Using High-Resolution Mass Spectrometry"

_ijms, 2022, doi:10.3390/ijms23073585_

Round 1

Reviewer 1 Report

In this paper, the authors use labelling of the reducing terminal hydroxyl group of carbohydrates with 18O and exchange of hydroxylic hydrogen atoms with deuterium to aid analysis of carbohydrates. Ten carbohydrates are used; five monosaccharides with reducing termini, mannitol and some more complex carbohydrates. The reducing carbohydrates are shown to exchange the oxygen at the reducing terminus with 18O and this label is used to provide information on the structures of fragment ions produced by collision-induced fragmentation (CID). For exchange of hydroxylic hydrogens, a new inlet system to the mass spectrometer is presented and shown to exchange a number of the hydrogen atoms.

The experiments are well performed but, unfortunately, the significance of the work is not clearly stated. Both types of exchange reaction have already been used by other investigators (several relevant papers are cited) over many years and the authors of this paper do not state clearly how their work advances the field or aids carbohydrate analysis.

Reaction with 18O is shown to result mainly in exchange of one oxygen atom through the open-chain form of reducing monosaccharides. Also observed after lengthy incubations are small amounts of the products of a second exchange for which a reasonable mechanism is proposed. Maltohexaose was observed to incorporate a third oxygen atom for which a possible explanation is given as “an isomerization of one of the end monomers”. What does this mean? If two of the oxygen atoms are incorporated at the reducing end of the molecule, there is only one other end and it is difficult to see how this could isomerize. This section of the paper should be expanded.

The incorporation of 18O is used to aid the elucidation of fragmentation mechanisms following CID but the mechanisms are only taken as far as proposals as to which parts of the molecules are present in the various fragment ions. There is no confirmation of the composition of the ions other than that given by the 18O incorporation. For D-glucose, the losses proposed for the fragment ion at m/z 119 appear satisfactory. The ion at m/z 113/115 is not discussed but a proposed composition with only three oxygen atoms is hard to explain. Other ions appear to be of mixtures of structures as shown in Figure 4. m/z 73 is presumably formed from m/z 91 by loss of H2O. Of the several possible positions for this to occur, only eliminations from the sugar ring are shown. Elimination of the 6-hydroxyl group (O11) does not appear to have been considered. Why? Can the authors draw any conclusions from their H/D experiments as to the sources of the hydrogen atoms that are eliminated?

The difference in structure between m/z 91 and 71 appear to be only the presence or absence of the ring oxygen (O6). Yet this difference in mass is 20 units rather than the18 units expected. The mass of the ion labelled as m/z 71 is, in fact 73 mass units, accounting for the difference.

The authors have constructed a novel inlet system for performing H/D exchange and shown the production of a biphasic envelope of peaks from two of the reference carbohydrates, consistent with earlier results with a more conventional ion source. However, this is as far as the experiment goes; there is no comment as to how this observation aids carbohydrate analysis.

Minor points

Figure 1. What is the significance of the shading on one of the pairs of spectra? In some cases, it appears to highlight the unlabelled spectrum and in others, it covers the labelled spectrum following exchange.

Figure 3: The axis labels are much too small.

Figure 4: Presumably the masses are of the [M - H]- ions from 18O-labelled glucose. This should be stated. Also there should be a statement that it is the highlighted groups that are retained in the ion. Some of the lettering is much too small to read. Also the highlight on the OH groups (red on red) make the lettering almost impossible to read. It would be more useful if the standard Domon and Costello nomenclature for showing the proposed structures of these ions were shown.

The text contains a large number of typographical errors such as missing spaces between words (e.g. line 97, exchangereactions) or after periods, and miss-spelled words (e.g. line 44, sifted).

m/z should be in italics throughout the paper.

In line 139, spectrum (singular) should be used instead of spectra (plural).

Line 143, units required after 2.0042.

Line 200, reference needed.

In the References section, the 16 and 18 isotope designators before O and not as superscripts.

Reference 8, the authors’ names are in capitals.

Reference 9, Change Medicago Truncatula via… to Medicago truncatula via…. (note use of capitals and italics).

Reference 10, the journal name is entered twice, once abbreviated and once not abbreviated.

Reference 27, some of the authors’ names are incorrect.

There are a few possibly novel aspects to this paper, e.g. the observation of two exchanges of oxygen from reducing sugars and the new inlet system for performing H/D exchange. However, the paper would be much improved if the significance and use of these observations to carbohydrate analysis were expanded as implied in the Introduction, possibly with some examples. The authors should also carefully review the text and correct the many errors, a few of which are highlighted above.

Author Response

Dear Reviewer 1,

please find below our detailed respond to comments.

Open Review

English language and style

(x) Extensive editing of English language and style required
( ) Moderate English changes required
( ) English language and style are fine/minor spell check required
( ) I don't feel qualified to judge about the English language and style

Yes

Can be improved

Must be improved

Not applicable

Does the introduction provide sufficient background and include all relevant references?

( )

(x)

( )

( )

Is the research design appropriate?

(x)

( )

( )

( )

Are the methods adequately described?

(x)

( )

( )

( )

Are the results clearly presented?

( )

(x)

( )

( )

Are the conclusions supported by the results?

( )

( )

(x)

( )

Comments and Suggestions for Authors

In this paper, the authors use labelling of the reducing terminal hydroxyl group of carbohydrates with 18O and exchange of hydroxylic hydrogen atoms with deuterium to aid analysis of carbohydrates. Ten carbohydrates are used; five monosaccharides with reducing termini, mannitol and some more complex carbohydrates. The reducing carbohydrates are shown to exchange the oxygen at the reducing terminus with 18O and this label is used to provide information on the structures of fragment ions produced by collision-induced fragmentation (CID). For exchange of hydroxylic hydrogens, a new inlet system to the mass spectrometer is presented and shown to exchange a number of the hydrogen atoms.

The experiments are well performed but, unfortunately, the significance of the work is not clearly stated. Both types of exchange reaction have already been used by other investigators (several relevant papers are cited) over many years and the authors of this paper do not state clearly how their work advances the field or aids carbohydrate analysis.

Answer: We agree, additional references were added:

Beneito-Cambra M., Bernabé-Zafón V., Herrero-Martínez J.M.,  Ramis-Ramos G.  Study of the Fragmentation of D-Glucose and Alkylmonoglycosides in the Presence of Sodium Ions in an Ion-Trap Mass Spectrometer, Analytical Letters, 2009, 42:6, 907-921, DOI: 10.1080/00032710902721956

Domon, B., & Costello, C. E. A systematic nomenclature for carbohydrate fragmentations in FAB-MS/MS spectra of glycoconjugates. Glycoconjugate journal, 1988, 5(4), 397-409. DOI: https://doi.org/10.1007/BF01049915

Kostyukevich Y., Sosnin S., Osipenko S., Kovaleva O., Rumiantseva L., Kireev A., Zherebker A., Fedorov M., Nikolaev E.N. PyFragMS─A Web Tool for the Investigation of the Collision-Induced Fragmentation Pathways ACS Omega Article ASAP DOI: 10.1021/acsomega.1c07272

The text was corrected.

Reaction with 18O is shown to result mainly in exchange of one oxygen atom through the open-chain form of reducing monosaccharides. Also observed after lengthy incubations are small amounts of the products of a second exchange for which a reasonable mechanism is proposed. Maltohexaose was observed to incorporate a third oxygen atom for which a possible explanation is given as “an isomerization of one of the end monomers”. What does this mean? If two of the oxygen atoms are incorporated at the reducing end of the molecule, there is only one other end and it is difficult to see how this could isomerize. This section of the paper should be expanded.

Answer: Dear Reviewer, thank you for your comment. Indeed, the aldose-ketose transformation may be an explanation for the second exchange in maltohexaose. As maltohexaose is hardly and irreversibly hydrolyzed, the only reasonable explanation of the third incorporated 18O label is the exchange in hydroxyl group. We made corrections in the text.

The incorporation of 18O is used to aid the elucidation of fragmentation mechanisms following CID but the mechanisms are only taken as far as proposals as to which parts of the molecules are present in the various fragment ions. There is no confirmation of the composition of the ions other than that given by the 18O incorporation. For D-glucose, the losses proposed for the fragment ion at m/z 119 appear satisfactory. The ion at m/z 113/115 is not discussed but a proposed composition with only three oxygen atoms is hard to explain. Other ions appear to be of mixtures of structures as shown in Figure 4. m/z 73 is presumably formed from m/z 91 by loss of H2O. Of the several possible positions for this to occur, only eliminations from the sugar ring are shown. Elimination of the 6-hydroxyl group (O11) does not appear to have been considered. Why? Can the authors draw any conclusions from their H/D experiments as to the sources of the hydrogen atoms that are eliminated?

Answer: Unfortunately we didn’t succeed in obtaining high quality CID spectra for molecules after H/D exchange. Of course, such data may help in the understanding of the fragmentation pathways, however, it is still difficult to account for the intramolecular rearrangement and hydrogen transfer. We are planning to it in future research.  Following text was added:

Previously, the number of research was dedicated to the investigation of the oligosaccharides using tandem mass spectrometry. However, less attention has been paid to the CID spectra of monosaccharides and their derivatives26,27.

The difference in structure between m/z 91 and 71 appear to be only the presence or absence of the ring oxygen (O6). Yet this difference in mass is 20 units rather than the18 units expected. The mass of the ion labelled as m/z 71 is, in fact 73 mass units, accounting for the difference.

Answer: The mass difference corresponds to the H218O. Currently PyFragMS suggests to user a fragments which theoretical molecular formulas differs from the experimental by <2 hydrogens. The reason is that we cannot account for all possible sites of protonation/deprotonation, intramollecular rearrangement and hydrogen migration.   

The authors have constructed a novel inlet system for performing H/D exchange and shown the production of a biphasic envelope of peaks from two of the reference carbohydrates, consistent with earlier results with a more conventional ion source. However, this is as far as the experiment goes; there is no comment as to how this observation aids carbohydrate analysis.

Answer: We agree. The text was revised as follows:

“Also, we can see, that the proposed design of the ion source allows the observation of the large number of exchanges, making it possible for distinguishing carbohydrate ions in complex mixtures. Indeed, when analyzing a mixture of unknown molecules with high molecular mass it may be difficult even to determine the class of the molecule, especially if the fragmentation pattern of the molecule is poor (as it is in the case of carbohydrates). However, the H/D exchange experiment can determine the number of –OH groups in the molecule and carbohydrates can be easily distinguished from other classes of molecules by the presence of large number of fast exchangeable hydrogens. The important advantage of the proposed method is its independence of the type of ionization (protonation, deprotonation or cationization with metal) and possibility to be coupled to liquid chromatography.  ”

Minor points

Figure 1. What is the significance of the shading on one of the pairs of spectra? In some cases, it appears to highlight the unlabelled spectrum and in others, it covers the labelled spectrum following exchange.

Answer: corrected.

Figure 3: The axis labels are much too small.

Answer: corrected.

Figure 4: Presumably the masses are of the [M - H]- ions from 18O-labelled glucose. This should be stated. Also there should be a statement that it is the highlighted groups that are retained in the ion. Some of the lettering is much too small to read. Also the highlight on the OH groups (red on red) make the lettering almost impossible to read. It would be more useful if the standard Domon and Costello nomenclature for showing the proposed structures of these ions were shown.

Answer: Figure 4 and its capture were corrected

The text contains a large number of typographical errors such as missing spaces between words (e.g. line 97, exchange reactions) or after periods, and miss-spelled words (e.g. line 44, sifted).

Answer: corrected.

m/z should be in italics throughout the paper.

Answer: corrected.

In line 139, spectrum (singular) should be used instead of spectra (plural).

Answer: corrected.

Line 143, units required after 2.0042.

Answer: corrected.

Line 200, reference needed.

Answer: corrected.

In the References section, the 16 and 18 isotope designators before O and not as superscripts.

Answer: corrected.

Reference 8, the authors’ names are in capitals.

Answer: corrected.

Reference 9, Change Medicago Truncatula via… to Medicago truncatula via…. (note use of capitals and italics).

Answer: corrected.

Reference 10, the journal name is entered twice, once abbreviated and once not abbreviated.

Answer: corrected.

Reference 27, some of the authors’ names are incorrect.

Answer: corrected.

There are a few possibly novel aspects to this paper, e.g. the observation of two exchanges of oxygen from reducing sugars and the new inlet system for performing H/D exchange. However, the paper would be much improved if the significance and use of these observations to carbohydrate analysis were expanded as implied in the Introduction, possibly with some examples. The authors should also carefully review the text and correct the many errors, a few of which are highlighted above.

Submission Date

27 February 2022

Date of this review

09 Mar 2022 12:09:20

Reviewer 2 Report

The authors have reported a preliminary, basic study on the ability of individual carbohydrates to undergo the H/D-exchange and 16O/18O-exchange reactions. They observed the excepted exchange in the carbonyl oxygen and an unexpected second exchange which occurs in hydroxyl groups. They gave probable explanation for the latter. The authors have proposed the fragmentation pathways for the 18-labeled D-glucose. They studied the H/D exchange reaction for different oligosaccharides using a modified ESI ion source. The manuscript has high novelty, significance, and scientific quality. The subject of this paper should be of interest for the readers of the Journal, the manuscript is acceptable for publication.

Minor comments:

Line 68. Please replace the Russian “i” with “and”

Line 114. D-arabinose is listed twice, but D-mannose and D-rhamnose are missing.

Figure 3. The mass of C3H3[18O]O may be wrong (73 not 71)

Line 148. Please use LC-MS/MS (not LC-MSA/MS)

Line 161. The word “labled” may be misspelled.

Line 175. Please correct “m//z”

Line 182. The word “intamolecular” is misspelled.

Line 192. Figure 5B (not 3B)

Line 205. “sof” is misspelled.

In “Materials and Methods”, the collision energy of the MS/MS experiments should be specified.

Author Response

Dear Reviewer 2,

please find below our detailed respond to comments.

Open Review

English language and style

( ) Extensive editing of English language and style required
( ) Moderate English changes required
(x) English language and style are fine/minor spell check required
( ) I don't feel qualified to judge about the English language and style

Yes

Can be improved

Must be improved

Not applicable

Does the introduction provide sufficient background and include all relevant references?

(x)

( )

( )

( )

Is the research design appropriate?

(x)

( )

( )

( )

Are the methods adequately described?

(x)

( )

( )

( )

Are the results clearly presented?

(x)

( )

( )

( )

Are the conclusions supported by the results?

(x)

( )

( )

( )

Comments and Suggestions for Authors

The authors have reported a preliminary, basic study on the ability of individual carbohydrates to undergo the H/D-exchange and 16O/18O-exchange reactions. They observed the excepted exchange in the carbonyl oxygen and an unexpected second exchange which occurs in hydroxyl groups. They gave probable explanation for the latter. The authors have proposed the fragmentation pathways for the 18-labeled D-glucose. They studied the H/D exchange reaction for different oligosaccharides using a modified ESI ion source. The manuscript has high novelty, significance, and scientific quality. The subject of this paper should be of interest for the readers of the Journal, the manuscript is acceptable for publication.

Answer: Dear reviewer, we are thankful for the comments.  The paper was improved as suggested.

Minor comments:

Line 68. Please replace the Russian “i” with “and”

Answer: corrected.

Line 114. D-arabinose is listed twice, but D-mannose and D-rhamnose are missing.

Answer: corrected.

Figure 3. The mass of C3H3[18O]O may be wrong (73 not 71)

Answer: corrected.

Line 148. Please use LC-MS/MS (not LC-MSA/MS)

Answer: corrected.

Line 161. The word “labled” may be misspelled.

Answer: corrected.

Line 175. Please correct “m//z”

Answer: corrected.

Line 182. The word “intamolecular” is misspelled.

Answer: corrected.

Line 192. Figure 5B (not 3B)

Answer: corrected.

Line 205. “sof” is misspelled.

Answer: corrected.

In “Materials and Methods”, the collision energy of the MS/MS experiments should be specified.

 Answer: corrected.

Submission Date

27 February 2022

Date of this review

03 Mar 2022 10:59:13

Round 2

Reviewer 1 Report

The authors have answered all of the points raised in the first review and made the necessary corrections.